

**The value of water isotope data on improving process**
**understanding in a glacierized catchment on the Tibetan**
**Plateau**
Yi Nan[1], Lide Tian[2,3], Zhihua He[4], Fuqiang Tian[1], Lili Shao[2]
[1]Department of Hydraulic Engineering, State Key Laboratory of Hydroscience and Engineering,
Tsinghua University, Beijing 100084, China
[2]Institute of International Rivers and Eco-security, Yunnan University, Kunming, China
[3]CAS Center of Excellence in Tibetan Plateau Earth Sciences, Beijing 100101, China
[4]Center for Hydrology, University of Saskatchewan, Saskatchewan, Canada
*Corresponding to*: Fuqiang Tian
Address: Room 330 New Hydraulic Building, Tsinghua University, Beijing 100084, China
Email: tianfq@mail.tsinghua.edu.cn



**Abstract**
This study integrated a water isotope module into the hydrological model THREW which has been
successfully used in high and cold regions. Signatures of oxygen stable isotope ($^{18}$O) of different water
inputs and stores were simulated coupling with the simulations of runoff generations. Isotope
measurements of precipitation water samples and global precipitation isotope product, as well as assumed
constant isotope signature of ice meltwater were used to force the isotope module. Isotope signatures of
water stores such as snowpack and subsurface water were updated by an assumed completely mixing
procedure. Fractionation effects of snowmelt and evapotranspiration were modeled in a Rayleigh
fractionation approach. The isotope-aided model was subsequently applied for the quantifications of
runoff components and estimations of mean water travel time (MTT) and mean residence time (MRT) in
the glacierized watershed of Karuxung River on the Tibetan Plateau. Model parameters were constrained
by three different combinations of observations including a single-objective calibration using streamflow
measurement solely, a dual- objective calibration using both streamflow measurement and MODIS
estimated snow cover area, and a triple- objective calibration using additionally isotopic composition of
stream water. Modeled MTT and MRT was validated by estimate of a tracer-based sine-wave method.
Results indicate that: (1) the proposed model performed quite well on simultaneously reproducing the
observations of streamflow, snow cover area, and isotopic composition of stream water, despite that only
precipitation water samples were available for tracer input; (2) isotope data helped to estimate more
plausible contributions of runoff components (CRCs) to streamflow in the melting season, and improved
the robustness of MTT and MRT estimations; (3) involving isotope data for the model calibration
obviously reduced uncertainties of the quantification of CRCs and estimations of MTT and MRT, through
better constraining the strong competitions among different runoff processes induced by meltwater and
rainfall. Our results inform high value of water isotope data on improving process understanding in a
glacierized basin on the Tibetan Plateau.
**Keywords:**
Tracer-aided hydrological model; Contribution of runoff components; Water travel time; Glaciered
catchment; Tibetan Plateau






## 1. Introduction


The Tibetan Plateau as a high mountainous cryosphere is the source of many major rivers in Asia
including Yarlung Tsangpo-Brahmaputra River, Ganges River, Indus River and so on (Report of STEP,
2018). Scientific understanding of hydrological processes in this region is critical in predicting the
responses of water resources and water hazards to climate changes (Lutz et al., 2014; Immerzeel et al.,
2010; Miller et al., 2012). River runoff in these basins is prominently fed by multiple water sources
including snowmelt, glacier melt and rainfall (Li et al., 2019). Coupling with the strong spatio-temporal
variabilities of meteorological inputs, the complicated runoff generation processes imply big challenges
in understanding the hydrological behaviors in glacierized basins on the Tibetan Plateau.
It is, therefore, of critical importance to quantify contributions of runoff components (CRCs) to
streamflow in glacierized regions. Estimating CRCs by hydrological models is one of the commonly
adopted method (Weiler et al. 2018), which is particularly subject to the following challenges. First,
modeled CRCs rely heavily on the model conceptualizations of the mixing and propagations of different
water sources in the basin. Model configurations and corresponding parameters representing the storage
capacities of soil layers and groundwater aquifers obviously affect the relative proportions of surface and
subsurface flow to streamflow (Nepal et al. 2014). CRCs modeled by different hydrological models are
thus rarely comparable (Tian et al. 2020). For example, Nepal et al. (2015) and Siderius et al. (2013)
compared CRCs estimated by different glacio-hydrological models in glacierized basins in the
Himalayan region, and demonstrated considerable variations of the modeled CRCs. They attributed the
difference to the variations of the model conceptualizations. Second, strong compensatory effects of the
simulated runoff induced by precipitation and ice meltwater which were typically not well constrained
in the model resulted in large variations of the modeled CRCs. For instance, modeling results from
Duethmann et al. (2015) and Finger et al. (2015) indicated that overestimated precipitation-triggered
runoff in the model can be easily compensated by an underestimated ice melt runoff and vice versa,
especially in high altitude glacierized basins where precipitation input have large uncertainty.
Tracer data of water stable isotope have been widely used to label runoff components in the popular
end-member mixing approach (e. g., Kong et al. 2011; He et al. 2020). Its value for improving modeled
CRCs, however, have not been sufficiently investigated. Previous applications of tracer-aided
hydrological models which integrated the simulation of water isotopic compositions of different runoff
components into the rainfall/melting-runoff processes in snow dominated basins have demonstrated high
values of water isotope data on diagnostically improving model structure and recognizing the dominances
of runoff processes on streamflow (Capell et al., 2012; Delavau et al. 2017; Son and Sivapalan, 2007;
Birkel et al., 2011; Stadnyk and Holmes, 2020). An early test of the isotope-aided hydrological model in
a glacierized basin in Tianshan Central Asia of He et al. (2019) indicated that additionally use of isotope
data helped to constrain the internal apportionments of runoff components in the model and improved
the estimation of CRCs at an event scale. However, exploring the values of water isotope data for
hydrological modelling in glacierized basins are still limited to the low availability of water tracer data
from field water sampling due to the harsh environment, especially for glacierized basins on the Tibetan
Plateau. As far as we know, glacio-hydrological model coupled with the simulations of isotope signatures
have not been developed and tested in the Tibetan Plateau yet.
Quantifying the time from entrance of water to its exits is fundamental to understandings of flow
pathways and the storage and mixing processes (McGurie and McDonnell, 2006). Characterizing water
travel time distribution (TTD) and mean travel time (MTT) in addition to the traditional focus on



streamflow response allows us to be closer to getting the right answers for the right reasons (Hrachowitz
et al. 2013). Despite that TTD and MTT serve good tools to diagnose unsuitable model structures and
parameterizations (McMillan et al. 2012), it has been rarely quantified in glacierized basins. Plenty of
convenient tools have been developed based on lumped parameter models, but their practical applications
in glacierized basins are restricted by the time invariant assumption and the weakness on considering the
strong spatio-temporal variability of runoff processes (van Huijgevoort et al. 2016) as well as the seasonal
water inputs from snowmelt and glacier melt. Fully physically-based water particle tracking approaches
coupling with hydrological processes whereas are only limited to small basins due to the heavy
computation cost (Remondi et al. 2018). Conceptual models that used additional tracer storage
compartments along with the flow and transport processes have provided crucial information on the
dynamics of flow pathways and storages, but rely heavily on the prior definitions of function shape (e.g.,
travel time distribution (TTD) in van der Velde et al. 2015; StorAge Selection function (SAS) in Benettin
and Bertuzzo 2018; age-ranked storage-discharge relation in Harman 2019). In contrast, tracer-aided
hydrological models that integrated the storage and transportation of conservative water tracers into the
runoff generation processes have been demonstrated as successful on estimating TTD and water ages as
well as their time variances with in snowmelt influenced basins (e.g., Soulsby et al. 2015; Ala-Aho et
al.2017). However, such hydrological models have not been applied in glacierized basins for estimations
of TTD and MTT yet.
For process understanding in glacierized basins, glacio-hydrological models that additionally
represented the snow processes and glacier evolution have been widely used (e.g, Immerzee et al. 2013;
Lutz et al. 2014 and 2016; Luo et al. 2018). The more complex integration of water sources from different
flow pathways and units whereas resulted in expanded parameter space of these hydrological models
which introduced large uncertainty in the model calibration (Finger et al. 2015). Equifinality is serious
in these regions when calibrating hydrological model by streamflow solely, indicating that different
parameters and runoff component proportions could perform similarly in discharge simulation (Beven
and Freer, 2001; Chen et al., 2017), despite of the general good performance for streamflow simulation.
Therefore, multiple datasets including glacier observation and remotely sensed snow products have been
frequently used in addition to streamflow measurements in vast glacio-hydrological simulations (e.g.,
Parajka and Blöschl, 2008; Konz and Seibert, 2010; Schaefli and Huss 2011; Duethmann et al. 2014;
Finger et al., 2015; He et al. 2018). However, both discharge and snow/glacier measurements provide
insufficient constraints on distributions of flow pathways and the parameterizations of subsurface water
storages (He et al. 2019). Although application in a glacierized basin in Central Asia of He et al. (2019)
indicated high utility of isotope data on constraining the complex interactions of multiple runoff
processes for the quantifications of CRCs, the values of water tracer such as stable isotope on reducing
uncertainties on the estimations of TTD and MTT in glacierized basins on Tibetan Plateau have not been
investigated.
In light of those backgrounds, this study integrated the simulation of oxygen isotope signatures into
a hydrological model that has been proved effective to simulate the runoff processes on the Tibetan
Plateau. The developed tracer-aided hydrological model was applied to the Karuxung River catchment
(286 km$^2$, 4550 to 7206 m a.s.l.) on Tibetan Plateau. The objectives of this study are: (1) to test the
capability of the proposed tracer-aided model on simultaneously reproducing streamflow and isotope
signatures of stream water in the study basin where only precipitation water samples are available for
isotope input, (2) to evaluate the values of tracer-aided method on improving the estimation of CRCs and
TTD/MTT in the study basin, and (3) to assess and interpret the differences between modeled TTD/MTT





and estimates by a lumped parameter method.

## 2. Materials and methodology

### 2.1 Study area and data

This study focuses on the Karuxung catchment, which is located in the upper region of the Yarlung
Tsangpo River basin, on the northern slope of the Himalayan Mountains (Figure 1). Digital elevation
model (DEM) data in the study catchment with a spatial resolution of 30-m was downloaded from the
Geospatial Data Cloud (www.gscloud.cn). The Karuxung river originates from the Lejin Jangsan Peak
of the Karola Mountain at 7206 m above sea level (a.s.l.), and flows into the Yamdrok Lake at 4550m
a.s.l. (Zhang et al., 2006). The catchment covers an area of 286 km$^2$. The river discharge is significantly
influenced by the headwater glaciers which cover an area of around 58 km$^2$ (Mi et al., 2001). This
catchment is dominated by a semi-arid climate. The mean annual temperature and precipitation at
Langkazi Weather Station were 3.4°C and 379 mm, respectively. Due to the effect of the South Asian
Monsoon, more than 90% of the annual precipitation falls between June and September. Precipitation
occurs mostly in form of snow from October to the following March at high elevations (Zhang et al.,
2015).

**[Figure 1]**
Daily temperature and precipitation data from 1$^{st}$ January 2006 to 30$^{th}$ September 2012 were
collected at the Langkazi Weather Station (4432 m a.s.l.). Altitudinal distributions of temperature and
precipitation across the catchment were estimated by the lapse rates reported in Zhang et al. (2015).
Runoff were measured daily from 1$^{st}$ April 2006 to 31$^{st}$ December 2012 at the Wengguo Hydrological
Station at the catchment outlet. The coverages of glaciers were extracted from the Second Glacier
Inventory Dataset of China (Liu, 2012). The 8-day snow cover extent data from MODIS product of
MOD10A2 (500m×500m, Hall and Riggs, 2016) were used to denote the fluctuations of the snow cover
area (SCA). The 8-day Leaf Area Index (LAI) and the monthly normalized difference vegetation index
(NDVI) data were downloaded from MODIS product of MOD15A2H (500m×500m, Myneni et al., 2015)
and MOD13A3 (1km×1km, Didan, 2015). Soil hydraulic parameters were estimated based on the soil
properties extracted from the 1km × 1km Harmonized World Soil Database (HWSD,
http://www.fao.org/geonetwork/).
Grab samples of precipitation and stream water were collected at the Wengguo Station in 2006-
2007 and 2010-2012, for analysis of δ$^{18}$O and δ$^2$H, and the characteristics of samples are summarized in
Table1. In the dry seasons when precipitation water was not sampled due to small event amounts,
precipitation isotope data from monthly Regionalized Cluster-based Water Isotope Prediction (RCWIP
with a pixel size of 10′×10′, Terzer et al., 2013) were used as proxy for model input. The effect of
elevation on the isotopic composition of precipitation was estimated using a lapse rate of -0.34‰/100m
based on Liu et al. (2007). The stream water samples were collected weekly every Monday from the river
channel near the Wengguo Station. Isotopic composition of glacier meltwater was assumed to be constant
during the entire study period and the value reported in Gao et al. (2009) was adopted.
**[Table 1]**

### 2.2 Tracer-aided hydrological model

The THREW (Tsinghua Representative Elementary Watershed) model was originally developed by



Tian et al. (2006), and has been successfully applied to a wide range of catchments (e.g., Tian et al., 2012;
Yang et al., 2014), including glacierized basins in the Alps, Tianshan, and the Tibet Plateau (He et al.,
2014; 2015; Xu et al., 2019). The THREW model uses the Representative Elementary Watershed (REW)
method for the spatial discretization of catchment, in which the study catchment is divided into REWs
based on the catchment DEM, and then each of the REWs is divided into sub-zones as the basic units for
hydrological simulation. More details of the model set up are given in Tian et al. (2006). In this study,
the Karuxung catchment was divided into 41 REWs.

Meltwater from snow and glacier are simulated using a temperature-index method as given in Eqs.
(1) and (2):
$$M_N = \begin{cases} DDF_N * (T - T_{N0}) & for\ T > T_{N0} \\ 0 & for\ T \leq T_{N0} \end{cases} \tag{1}$$

$$M_G = \begin{cases} DDF_G * (T - T_{G0}) & for\ T > T_{G0} \\ 0 & for\ T \leq T_{G0} \end{cases} \tag{2}$$

where, the subscripts $N$ and $G$ represent snow and glacier, respectively. $M$ is the melt amount, $T$ is
temperature and $T_0$ refers to temperature threshold above which snow/ice starts to melt. $DDF$ is the
degree-day factor, representing the melt rate. Glacier meltwater ($M_G$) in this study includes both ice melt
and snowmelt on the glacierized area.

The fraction of snowfall ($P_N$) of the total precipitation $P$ is determined by a temperature threshold
$T_N$ in Eq. 3. Snow water equivalent (SWE) of each REW is thus updated by Eq. 4. The snow cover area
(SCA) of the corresponding REW is determined by a SWE threshold value ($SWE_0$): when the calculated
SWE is higher than $SWE_0$, the SCA of this REW is recorded as 1, otherwise the SCA is assumed to be 0
(similarly to Parajka and Blöschl, 2008; Zhang et al., 2015; He et al., 2014). The SCA of the whole study
catchment is calculated as the ratio of the sum of the areas of snow covered REWs to the total catchment
area. Values of $T_N$ and $SWE_0$ are set based on prior knowledge from Dou et al. (2011), Marques et al.
(2011) and He et al. (2014): $T_N$ = 2°C, $SWE_0$ = 20mm.
$$P_N = \begin{cases} 0 & T \leq T_N \\ P & T > T_N \end{cases} \tag{3}$$

$$\frac{dSWE}{dt} = P_N - M_N \tag{4}$$

Meltwater of ice and snow, and rainfall over the glacier area are assumed to flow directly into the
channel near the glacier tongue in form of surface runoff, based on the low permeability of the glacier
surface. Snowmelt in the non-glacier area is assumed to generate runoff in a similar way to rainfall
(Schaefli et al., 2005). For model simplicity, the evolution of the glacier area is not simulated in the model
for the short simulation period of five years.

Simulation of δ18O of multiple water sources was integrated into the runoff generation processes
in the THREW model (hereafter abbreviated as a THREW-t model). The δ18O of water sources in each
of the sub-zones was assumed to be conservative, meaning that no chemical reactions occurred during
the mixing of water sources. We assumed that the isotopic compositions of precipitation and glacier
meltwater are linearly dependent on elevation, and used linear gradients reported in Liu et al. (2007)
to estimate the initial isotopic compositions of precipitation and glacier meltwater in individual REWs
(similarly to He et al. 2019). The isotopic compositions of the snowpack and subsurface water storages





were initialized by a "spin-up" running for three hydrological years, assuming the isotopic
compositions of water storages would reach steady levels after three years' running. Isotope
composition of event snowfall on the snowpack was assumed to be the same as that of precipitation
occurring in the corresponding REW.

The fractionation effects of evaporation on isotope composition of water were estimated by a
Rayleigh fractionation method in Eqs. (5) to (7) (Hindshaw et al., 2011; Wolfe et al., 2007; He et al.,
2019):

$$\delta^{18}O_x{'} = \delta^{18}O_x * \frac{1-f^{CF\left(\frac{1}{\alpha}-1\right)+1}}{1-f} \qquad (5)$$

$$ln\alpha = -0.00207 + \frac{-0.4156}{T} + \frac{1137}{T^2} \qquad (6)$$

$$f = 1 - \frac{w_x'}{w_x} \qquad (7)$$

where, $\delta^{18}O_x{'}$ is the isotope composition of the evaporated water, $\delta^{18}O_x$ is the isotope composition of
water before evaporation, $\alpha$ is the Rayleigh fractionation factor, $T(K)$ is air temperature in the
corresponding catchment unit, $CF$ is a correction factor, and $f$ is the ratio of remaining water volume to
the original water volume before evaporation.

A complete mixing assumption was used for the tracer signatures in each water storage.
Consequently, $\delta^{18}$O of soil water and groundwater were updated according to the following equation:
$$\delta^{18}O_t = \frac{w_o \delta^{18}O_o + \sum w^i \delta^{18}O^i}{w_o + \sum w^i} \qquad (8)$$

where, $w_o$ and $\delta^{18}O_o$ are the water quantity and isotopic composition of the subsurface storages at the
prior step, respectively. $w^i$ refers to the infiltration into the soil storage from water source $i$. For
groundwater storage, $w^i$ refers to the seepage from upper soil water. $\delta^{18}O^i$ stands for the isotopic
composition of input water source $i$.

Stream water in each of the REWs was considered as a mixture of three components including
inflow from the upstream REWs, runoff generated in the current REW, and the water storage in the
river channel. Consequently, the isotopic composition of stream water in each REW ($\delta^{18}O_r$) was
estimated based on the following conservative mixing equation:
$$\delta^{18}O_r = \frac{\delta^{18}O_{r0} * w_r + \sum \delta^{18}O_{r,up}{}^k * I^k + \delta^{18}O_{sur}R_{sur} + \delta^{18}O_{gw}R_{gw}}{w_r + \sum I^k + R_{sur} + R_{gw}} \qquad (9)$$

where, $\delta^{18}O_{r_0}$ is the isotopic composition of stream water and $w_r$ is the water storage in the river
channel at the time step before the mixing of runoff components. $\delta^{18}O_{r,up}{}^k$ is the isotopic composition
of stream water coming from the upstream REW $k$, and $I^k$ is the inflow from the corresponding
upstream REW. Subscripts of $sur$ and $gw$ refer to the surface runoff and subsurface flow from
groundwater outflow generated in the current REW.
**2.3 Model calibration**

The physical meaning and value ranges of the calibrated parameters in the THREW-t model are
described in Table 2. Parameter values were optimized using three calibration variants: (1) single-



objective calibration using only the observed discharge at the catchment outlet, (2) dual-objective
calibration using both observed discharge and MODIS SCA estimates, and (3) triple-objective calibration
using observed discharge, MODIS SCA estimates and $\delta^{18}O$ measurements of stream water. Considering
the data availability, we chose April 1st 2006 to December 31st 2010 as the calibration period, and January
1st 2011 to September 30th 2012 as the validation period. For SCA, we used only the MODIS SCA
estimates during the ablation period (1st May to 30th July) of each year for the model calibration, because
simulations of runoff processes are mostly sensitive to the dynamics of snow cover extent in the melting
period (Duethmann et al., 2014). Only the $\delta^{18}O$ measurements of stream water in the rainy season (from
the first rainfall event to the last rainfall event of each year, as shown in Table 1) were used to optimize
the model parameters, because the measured isotope data for precipitation were only available in this
season. We chose the objective functions of Nash Sutcliffe efficiency coefficient (NSE) (Nash and
Sutcliffe, 1970) and mean absolute error (MAE) to optimize the simulations of discharge, SCA and
isotope respectively (Eqs. 12-14).
$$NSE_{dis} = 1 - \frac{\sum_{i=1}^{n}(Q_{o,i} - Q_{s,i})^2}{\sum_{i=1}^{n}(Q_{o,i} - \overline{Q_o})^2} \tag{10}$$

$$MAE_{SCA} = \frac{\sum_{i=1}^{n}|SCA_{o,i} - SCA_{s,i}|}{n} \tag{11}$$

$$MAE_{iso} = \frac{\sum_{i=1}^{n}|\delta^{18}O_{o,i} - \delta^{18}O_{s,i}|}{n} \tag{12}$$

where, $n$ is the total number of observations. Subscripts of $o$ and $s$ refer to observed and simulated
variables, respectively. $\overline{Q_o}$ is the average value of observed streamflow during the assessing period.
An automatic procedure based on the pySOT optimization algorithm developed by Eriksson et al.
(2015) was implemented for all the three calibration variants to identify the behavioral parameters.
pySOT used surrogate model to guide the search for improved solutions, with the advantage of requiring
few function evaluations to find a good solution. An event-driven framework POAP were used for
building and combining asynchronous optimization strategies. The optimization was stopped if a
maximum number of allowed function evaluations was reached, which was set as 3000 in this study. For
the single-, dual- and triple-objective calibration variants, $NSE_{dis}$, $NSE_{dis}$ - $MAE_{SCA}$, $NSE_{dis}$ - $MAE_{SCA}$ -
$MAE_{iso}$ were chosen as combined optimization objectives, respectively. The pySOT algorithm was
repeated 150 times for each calibration variant. The 150 final results were further filtered according to
the metric of $NSE_{dis}$, i.e., only the parameters producing $NSE_{dis}$ higher than a threshold were regarded as
behavioral parameter sets. For single- and dual-objective calibration, the threshold was selected as 0.75.
Considering the trade-off between discharge and isotope simulation, the threshold was chosen as 0.70
for triple-objective calibration. For each calibration variant, the parameter producing highest combined
optimization objective was regarded as the best parameter set.

**[Table 2]**

**2.3 Quantifications of the contributions of runoff components to streamflow**
The contributions of individual runoff components to streamflow were quantified based on two
definitions of the runoff components. In the first definition, we quantified the contributions of individual
water sources including rainfall, snow meltwater and glacier meltwater to the total water input, which
were commonly reported in previous quantifications of runoff components on the Tibetan Plateau (Chen





et al., 2017; Zhang et al., 2013). To be noted, the sum of the three water sources should be larger than the
simulated volume of runoff because of the evaporation loss. Thus, contributions quantified in this
definition only refer to the fractions of the water sources in the total water input forcing runoff processes,
rather than the actual contributions of water sources to streamflow at the basin outlet. In the second
definition, runoff components were quantified based on the runoff generation processes including surface
runoff and subsurface flow. Surface runoff consists of runoff triggered by rainfall and meltwater that feed
streamflow through surface paths, and the precipitation occurring in river channel and contributes to
runoff directly. Subsurface flow is the interflow from groundwater outflow.
**2.4 Estimation of the water travel time and residence time**
In this study, the water travel time is estimated by three methods, a lumped analytical method and
two distributed model-based methods. A simplified lumped method, sine-wave method (SW) was used
to provide a reference value of mean travel time (MTT) and mean residence time (MRT) in the catchment.
The adopted model-based methods were developed by van Huijgevoort et al. (2016) and Remondi et al.
(2018), which were referred to as mass-mixing method (MM) and flux-tracking method (FT),
respectively. SW method is based on the isotope data of precipitation and stream water. MM and FT
methods were conducted by the tracer-aided hydrological model using behavior parameter values
identified by the calibration scenarios.
SW method has a stationarity assumption that a constant flow field gives constant travel time
distribution (TTDs) (van der Velde et al., 2015). It assumes the form of TTD, and derives the MTT
directly from the series isotopic data (McGuire and McDonnell, 2006). Although the assumption is rather
stringent, SW is widely used in the studies when an approximate estimation of MTT is required (e.g.,
Kirchner, 2016; Garvelmann et al., 2017). Here we assumed the form of TTD as the exponential function,
and the MTT can be estimated according to Eqs. 13-14 (McGuire and McDonnell, 2006; Garvelmann et
al., 2017):
$$\delta_t = \bar{\delta} + A * \sin\left(\frac{2\pi}{365} * t + \varphi\right) \tag{13}$$

$$MTT = \frac{\sqrt{\left(\frac{1}{A_r/A_p}\right)^2 - 1}}{2\pi} \tag{14}$$

where, $\delta_t$ is the calculated $\delta^{18}O$ of stream water or precipitation on day $t$ of the year. $\bar{\delta}$ is the mean $\delta^{18}O$
of stream water or precipitation measured in different seasons. $A$ and $\varphi$ are parameters controlling the
amplitude and phase lag, and are estimated based on the fitness between the sine-wave curve and the
$\delta^{18}O$ measurements. Subscripts of $r$ and $p$ in Eq. 14 represent river and precipitation, respectively.
MM method was used to estimate the water age of outflow and water storage in the catchment. For
the outflow (e.g., stream water, evaporation), the concept of water age is consistent with the concept of
"travel time conditional on exit time" by Botter et al. (2011), "flux age" by Hrachowitz et al. (2013), and
"backward travel time" by Harman and Kim (2014). For the water storage (e.g., soil water, groundwater,
snowpack), the concept of water age is consistent with the concept of "residence time" by Botter at al.
(2011) and "residence age" by Hrochowitz et al. (2013). MM method regarded the water age as a kind
of tracer, and simulated the "concentration" of this tracer of the water bodies including snowpack, soil
water and stream water (van Huijgevoort et al., 2016; Ala-aho et al., 2017). The "mass" and



"concentration" of the water age were simulated similarly in Eqs. 8-9, by replacing $\delta^{18}O$ with water age
of the multiple terms. Event precipitation entering the catchment was treated as new water with a
youngest age equaling to the simulation step of model. The glacier meltwater was regarded as very old
water, and a constant age of 1000 days was adapted in this study. Meanwhile, the age of water stored in
snowpack, soil and river channel were assumed to increase with the ongoing simulation time: water age
increased by one day after each model running at a daily step.
FT method ran the model multiple times in parallel to track the fate of each precipitation event
separately (Remondi et al., 2018). All days with precipitation were individually labeled and tracked over
the simulation period by adding an artificial tracer to the water amounts which was assumed to not
otherwise exist anywhere. The snow meltwater was tracked from the time when the snow entered the
catchment as solid precipitation (i.e., snowfall), rather than the time when the snowpack melted. Glacier
meltwater was not tracked, because the evolution of glacier was not simulated in the model, and the travel
time of glacier melt as surface runoff was negligible. Similar as MM method, the MTT of glacier melt
runoff was assumed as a constant value as 1000 days. The mixing and transport processes of the tracer
were also simulated similarly in Eqs 8-9 by replacing $\delta^{18}O$ with the concentration of the artificial tracer.
By summarizing the mass of labeled precipitation in the water storage and stream water, the TTD
conditional on exit time (backward TTD), TTD conditional on injection time (forward TTD) and
residence time distribution (RTD) can be derived.
In summary, this study estimated the water travel time and residence time using a lumped method
(SW), and two model-based methods (MM and FT), and the results of three methods were compared to
test the robustness of travel time estimation in this glacierized basin. Specifically, SW method estimated
the MTT of total discharge and the MRT of water storage directly based on the isotopic data in stream
water and precipitation. MM method estimated the water age of stream water and groundwater storage,
representing the daily backward MTT and MRT respectively, and all the 19 behavioral parameter sets of
triple-objective calibration were used to illustrate the uncertainty of MTT. FT method estimated the time-
varying precipitation-triggered TTD and RTD, only using the parameter set producing best metric. To
make the result of FT method comparable to MM method, the glacier melt runoff was also assumed to
have MTT (water age) of 1000 days to calculate the MTT of the total runoff generation as the weighted
average value of the MTT of precipitation runoff (including rainfall and snowmelt) and glacier melt
runoff according to the contribution of water sources. The glacier melt was assumed to only contribute
to surface runoff directly and exit the catchment rapidly, thus had no influence on the MRT estimation.
**3. Results**
**3.1 Model performance on the simulations of discharge and isotopic composition**
For the calibration period, the single-objective calibration produced good performance for the
simulation of discharge, but had an extremely poor performance for the simulations of SCA and $\delta^{18}O$
(Table 3). Involving SCA in the calibration objective, the dual-objective calibration significantly
improved the simulation of SCA, and kept a good behavior on discharge simulation, but brought no
benefit to the isotope simulation. The triple-objective variant led to a good performance for all the three
metrics. The $NSE_{dis}$ produced by triple-objective calibration was slightly lower than that of another two
variants because of the lower threshold for behavior parameter sets. The simulation of isotopic
composition of stream water was significantly improved by triple-objective calibration compared to the
other two variants. For the validation period, the $NSE_{dis}$ of triple-objective calibration was significantly


improved, even better than the single-objective, indicating the improved process representation of the
behavior parameters by the triple-objective calibration. Through 150 runs of calibration program, triple-
objective calibration got the smallest behavior parameter sets, indicating that involving additional
calibration objectives increase the identifiability of model parameters and reduce the equifinality.
**[Table 3]**
Fig. 2 shows the uncertainty ranges of the simulations for the behavioral parameters obtained by
the three calibration variants. The three variants generally produced similar hydrographs in terms of the
magnitudes and timing of peak flows with averaged behavioral parameter sets, but the triple-objective
had a narrower uncertainty range, especially for the baseflow dominated periods (Figs. 2a-c). The single-
objective variant resulted in rather large uncertainty ranges for the simulations of SCA and isotopic
composition (Figs. 2d and g). The good fitness between the simulated and observed streamflow in
summer is likely due to the largely overestimated rainfall-triggered surface runoff, because of the
underestimated reduction of SCA in spring. The dual-objective calibration significantly reduced the
uncertainty range of the SCA simulation, and captured the declining SCA in summer very well (Fig. 2e).
Including SCA in the model calibration, however, only provided small benefits for the simulation of $\delta^{18}O$
in stream water (Fig. 2h). Simulations of the triple-objective variant properly reproduced the temporal
variation in SCA in the melt season, despite the slightly reduced performance compared to that of the
dual-objective variant (behaving as higher $MAE_{SCA}$ of triple-objective calibration in Table 3). Meanwhile,
the seasonal variations of $\delta^{18}O$ of stream water were reproduced well by the triple-objective calibration
(Fig. 2i).
**[Figure 2]**
Fig. 3a shows median value of the simulated daily inputs of water sources (rainfall, snowmelt, and
glacier melt) for the calibration period obtained by the behavioral parameter sets of the triple-objective
variant. All the three water sources started to contribute to stream water in around April. The volume of
snowmelt peaked around June, and then decreased rapidly in July as the catchment SCA decreased
significantly. The volumes of rainfall and glacier melt peaked in mid-summer which was the wettest and
warmest period in the year. The fluctuations of the simulated $\delta^{18}O$ of stream water in Fig. 3b are generally
consistent with the varying contributions of these water sources to runoff. At the beginning of the wet
season, $\delta^{18}O$ of stream water increased rapidly in response to the dominance of the isotopic enriched
precipitation. The $\delta^{18}O$ of stream water began to decease in the late wet season, likely because of the
reduced $\delta^{18}O$ of precipitation caused by the "temperature effect" (Dansgaard, 1964) and the effect of
southwest monsoon (Yin et al., 2006), as well as the increased contributions of isotopic depleted glacier
melt.
**[Figure 3]**
**3.2 Contributions of runoff components**
The results of runoff component quantification reported in this section were based on the behavioral
parameter sets of the three calibration variants. Table 4 and Figure 4 shows the proportions of water
sources in the mean annual water input during 1st January 2007 to 31st December 2011. In all the three
calibration variants rainfall provided most of the water quantity for runoff generation (44.2% to 48.0%),
because of the high partition of rainfall (around 347mm) in the annual precipitation (around 587mm).
The single-objective variant estimated the lowest proportion of snowmelt (19.7%), because the





simulation of SCA was not constrained in the calibration, leading to largely overestimated SCA in
comparison to the MODIS SCA estimates due to less melting (Fig. 2d). The dual-objective variant
estimated the highest proportion of glacier melt (33.8%), resulting in a lower proportion of rainfall
(44.2%). Involving the calibration objective of isotope, the triple-objective variant estimated the lowest
proportion of glacier melt (29.2%) by rejecting the parameter sets that produced high contribution of
glacier melt (as shown in Fig. 4), which will be discussed more in detailed in the discussion section. To
be noted, despite above differences, the results of three calibration variant were quite similar, with the
maximum difference lower than 5%. However, the uncertainties of the simulated water proportions
decreased substantially with the increase of data that was involved in the calibration, showing as a
decreasing uncertainty (12.4% to 6.2%, Table 4) and fewer outliers (Fig. 4), demonstrating considerable
values of additional datasets for constraining the simulations of corresponding runoff generation
processes.

**[Table 4]**

**[Figure 4]**

Fig. 5 compares the seasonal proportions of water sources in the total water input of the three
calibration variants. The seasonal dominance of the water sources on runoff estimated by the three
calibration variants are similar. In particular, the proportion of rainfall was large (around 55%) in summer
but small in winter when rainfall rarely occurred. Snowmelt and glacier melt dominated the total water
input in winter with proportions of around 60% and 40%, respectively. The proportion of meltwater in
summer was relatively low because of the dominance of rainfall during the summer monsoon. Snowmelt
could only account for around 15% in the total water input in summer because of the significantly reduced
snowpack. The proportion of glacier melt was higher than that of snowmelt in summer because of the
decreasing snow cover area. In the spring months, snowmelt and glacier melt contributed around 55%-
60% and 35%-30% to the total water input, respectively. Rainfall provided the remaining 5%. The glacier
melt provided a steady contribution of around 30%-40% throughout the entire hydrological year. The
seasonal proportions of water sources show slightly different among the calibration variants. Specifically,
the triple-objective calibration estimated not only the highest snowmelt and lowest glacier melt in the
three seasons except winter, but also the highest contribution of rainfall in summer (Fig. 5c). Single-
objective calibration produced the highest contribution of rainfall in autumn, and the highest contribution
snowmelt in winter (Fig. 5a). In addition, the uncertainty ranges of the seasonal proportion during
summer and autumn were obviously reduced by the triple-objective calibration (Fig. 5c).

**[Figure 5]**

Table 5 shows the contributions of runoff components to annual runoff. Three calibration variants
resulted in rather similar contributions of surface runoff and subsurface runoff (around 65% and 35%
respectively). Surface runoff was the dominant component in this catchment, because of the large glacier
covered area (around 20%) and the large saturation area (around 20%). The triple-objective calibration
estimated relatively lowest surface runoff (64.9%) and highest subsurface runoff (35.1%). Again, the
triple-objective calibration resulted in lowest uncertainty ranges for the contributions of the runoff
components compared to the other two calibration variants (4.1% compared to 12.1%). Isotope data used
in the triple-objective variant provided additional constraints on the estimation of parameters controlling
the generation of subsurface flow (such as *KKA* and *KKD* in Table 2) and the saturation area where
surface runoff occurred (such as *WM* and *B*), thus constraining the partitioning between surface runoff





and subsurface runoff.

**[Table 5]**

**3.3 Estimations of water travel time and residence time**

The travel time and residence time were estimated for the five water years (2007/1/1 – 2011/12/31)
during the simulation period. The result produced by the best parameter set ((NSE$_{dis}$ = 0.72, MAE$_{SCA}$ =
0.079, MAE$_{iso}$ = 0.484) was used to test the consistency between the two model-based methods. Based
on the assumption that glacier melt water had an age of 1000 days, the backward MTT and MRT
estimated by MM (FT) method were 1.70 (1.72) and 1.22 (1.17) years, respectively. Fig. 6 shows the
comparison between the results of MM and FT methods. As shown in Fig. 6a and 6d, there were strong
correlation between the daily MTTs (MRTs) estimated by the two methods with a high correlation
coefficient of 0.96 (0.98). The daily MTT and MRT series also showed similar temporal variability
between the two methods as shown in Fig. 6b and 6e. The MRT increased steadily during dry season,
and decreased rapidly during wet season due to the recharge of young precipitation. The daily MTT also
showed steady increasing trend during dry season, but showed significant fluctuation during wet season
because of the combined effect of young precipitation and old glacier meltwater. Fig. 6c and 6e shows
the probability density function of the daily backward MTT and MRT produced by the two methods. The
daily MTT had a large range from 0.42 to 2.75 years, with several peak density values at around 1, 1.5
and 2.67 years, including the influence of the multiple water sources with different ages. On the contrary,
the daily MRT only had a narrow range from 0.75 to 1.75 years, with a significant peak value at around
1.25 years, similar with the MRT. Excluding the effect of glacier meltwater, FT method estimated the
precipitation-triggered backward MTT of runoff as 263 days, significantly smaller than the MRT,
indicating the incomplete mixing in the catchment scale caused by the distributed modelling framework.

**[Figure 6]**

The lumped SW method estimated the MTT and MRT as 1.68 years (A$_r$ and A$_p$ were estimated as
0.58 and 6.19, respectively). Based on the average result of 19 behavioral parameter sets, the model-
based methods estimated the MTT and MRT as 1.61 and 1.28 respectively. The two kinds of methods
produced similar MTT, indicating the robustness of travel time estimation in this catchment. The
precipitation-triggered MTT (shorter than 1 year) was significantly smaller than the MTT of total runoff
estimated by the lumped method, indicating the effect of old glacier melt water. The glacier melt
contributed to stream water through surface runoff directly, and had no contribution to the water storage,
leading to a smaller model-based MRT compared to MTT. The uncertainty of MTT and MRT estimation
could be reflected by the range produced by MM method by the 19 behavioral parameter sets as shown
in Fig. 7. The standard deviation of the estimated MTT and MRT were 74 and 79 days, respectively. The
uncertainty range during June to August was relatively small (Fig. 7a), indicating that different behavioral
parameters produced similar precipitation-triggered processes during the wet season, and the uncertainty
mainly came from the large range of MRT, i.e., the age of water storage including soil water and
groundwater.

**[Figure 7]**

Based on the best parameter set, FT method tracked the transportation of precipitation and produced
time-varying forward TTD, backward TTD and RTD. For simplicity, Fig. 8 shows the average
distributions weighted by the precipitation amount (for forward TTD), runoff generation (for backward



TTD) and water storage (for RTD). As shown in Fig. 8a and 8b, the forward and backward TTD were
similar, behaving as a high proportion (~0.3) of the youngest water, which was consistent with the high
proportion of rapid surface runoff. The high proportion of young water led to a similar TTD form as
exponential model. The relative peaks of TTD were mainly around the travel time of integral years,
indicating the influence of baseflow from groundwater, which were recharged by precipitation in the wet
seasons of previous years. The simulated RTD was significantly different from the TTD, behaving as the
low probability density of young water. The young water was mainly recharged by the infiltrating rainfall
and snowmelt, which was negligible compared to the total water storage. Again, the difference between
TTD and RTD indicated the incomplete mixing processes, behaving as affinity for young water due to
the rapid flow pathways such as surface runoff.

**[Figure 8]**

**4. Discussions**
**4.1 The values of tracer on constraining flow pathways and water storages in the hydrological**
**model**

This study developed a tracer-aided hydrological model and tested its behavior in a glacierized
catchment. Because of the sampling difficulties on the Tibetan Plateau, the tracer data of the water
sources (e.g., snow, glacier, groundwater) was rather limited compared to other tracer-aided modelling
works (e.g., Ala-aho et al., 2017; He et al., 2019). Nonetheless, the model developed in this study
performed well on producing the tracer signature of stream water, producing a tool for applying the
tracer-aided method to the areas with limited tracer data. Although it was widely accepted that simple
input-output tracer measurements provided limited insight into catchment function, and sampling source
water components would be helpful (Birkel et al., 2014; Tetzlaff et al., 2014), the uncertainty of model
could still be reduced significantly by satisfying the output tracer signature (Delavau et al., 2017),
especially in cold regions where hydrological processes were more complex. The fact that the model can
simultaneously satisfy three calibration objectives over a long period gave confidence in the model
realizations (McDonnell and Beven, 2014).

Our results indicate that involving the isotope into calibration significantly reduced the uncertainty
of quantifying the runoff components. To understand the role of isotope data on reducing the uncertainty,
the results of dual-objective calibration variant were analyzed why some of the parameter sets behaved
poorly on isotope simulation despite their good performance on discharge and snow simulation. Among
the 117 behavioral parameter sets of dual-objective calibration, only 14 of them produced relatively good
isotope simulations ($MAE_{iso} < 1.0$). As shown in Fig. 9a and 9b, these 14 isotopic behavioral parameter
sets produced the proportion of runoff component within a relatively smaller range (27.5% to 38% for
glacier melt, and 58% to 75% for surface runoff), while the 117 behavioral parameter sets produced a
much larger variation (24% to 53% for glacier melt, and 40% to 90% for surface runoff). This indicated
that involving isotope data for model calibration helped to exclude some unreasonable proportions of
runoff component. The distribution of scatter in Fig. 9a and Fig. 9b was similar, and the proportion of
surface runoff had a strong correlation with the proportion of glacier melt as shown in Fig. 9c (because
of the assumption that glacier melt contributed to surface runoff directly), thus the mechanism that
isotope can reject unreasonable proportions were the same for water sources and runoff generation
processes. Fig. 9d shows the simulation range of $\delta^{18}O$ of stream water by calibrated parameters that
resulted in glacier melt proportion in the total water input higher than 40%. The simulated isotopic





signature showed strong fluctuations due to the high proportion of surface runoff with a larger time
variation compared to the relatively steady signature of subsurface runoff. Also, the simulated isotopic
values were significantly higher than the observations, which was mainly the result of the excessive
isotopic fractionation due to the too much evaporation of surface water (Hindshaw et al., 2011; Wolfe et
al., 2007). Fig. 9e shows the simulation range by the parameters with proportion of surface runoff lower
than 45%. In contrast to scenarios with too high glacier melt, the simulated isotope signature showed
small variation and the mean values were much lower than the observation. Our result also showed that
the proportion of surface runoff and glacier melt tended to be higher when the NSE$_{dis}$ was higher,
indicating that focusing on the simulation of integrated observation of discharge only will likely lead to
overestimated surface runoff and glacier melt. These results indicated that the isotope data helped to
constrain the quantifications of runoff components by (1) regulating the competition between rapid
component with strong variation of isotope signatures (e.g., surface runoff) and slow component with
relatively stable isotope signatures (e.g., subsurface runoff) to match the daily fluctuations of observed
isotope signature of stream water, and (2) controlling the isotopic fractionation by adjusting the
evaporation to satisfy the observed isotopic value.
**[Figure 9]**
Two model-based methods (MM and FT) were adopted to estimate travel time and residence time
in this study, and verified by the result of lumped method (SW). Both MM and FT methods can estimate
MTT and MRT, but FT provided more information including TTD and RTD, which was actually more
of interest. MM method has been used in several previous studies, including modelling work in snow-
influenced basins (Ala-aho et al., 2017). Consequently, the results of FT and MM were compared in this
study, to ensure that the additional information provided by FT method was reasonable. Our result
indicated that the two model-based methods produced consistent results, which were also similar with
the lumped method, indicating the robustness of MTT and MRT estimation through a tracer-aided model
without defining any prior distribution functions.
Although significantly constraining the proportion of runoff component, the uncertainty ranges of
simulated MTT and MRT, especially that during baseflow-dominant period (as shown in Fig. 7b) were
still rather large, indicating that the estimation of groundwater age had a large uncertainty, which was
similar with other model-based age estimation works (e.g., Ala-aho et al., 2017; van Huijgevoort et al.,
2016). The isotope observations were mainly collected during wet season when precipitation-triggered
surface runoff played an important role in runoff generation, thus this process was constrained relatively
well by the isotope calibration, showing as the similar fluctuation of MTT during wet season produced
by different parameter sets. Although the proportion of subsurface runoff was constrained, the storage
volume of groundwater was poorly constrained, because of the relatively simplified structure of the
groundwater module of THREW model (Tian et al., 2006), which adopted a two-layer reservoir model
to describe the processes of seepage and subsurface flow. Apart from involving more calibration
objectives, improving the physical mechanism and the representation of hydrological processes is another
important way to constrain the model behavior and reduce uncertainties.
**4.2 Limitation and uncertainty**
Multiple water sources brought difficulties to hydrological modelling in glacierized basins
(Zongxing et al., 2019). Focusing on the tracer transportation processes, the model developed in this
study made some simplifications on the processes related to snow and glacier to make the model structure





parsimonious. First, the snow accumulation and melting processes were simulated by a simple
temperature-based method, which was relatively lack of physical mechanism compared to the energy-
based methods (e.g., Pomeroy et al., 2007). Nonetheless, this method had an acceptable behavior and
was widely used in studies of snow simulation (e.g., He et al., 2014), and the simulated SCA was
validated by the MODIS data during ablation period in this study. Second, the evolutions of glacier
thickness and area were not simulated in the model. Simplification of a constant glacier area likely led
to an overestimation of the contribution of glacier melt to runoff, as the glacier cover area should get
smaller due to the climate warming. However, this simplification should have minor influence on the
result because the changes of glacier area was rather small in a short simulation period of seven years.
The lack of source water sampling made it difficult to fully validate the modelling result. Although
the isotope signature of stream water was reproduced well, it cannot guarantee that the isotopic variations
of groundwater, snowmelt were simulated correctly. The quantification of runoff components was also
hard to verified. The end-member method cannot be applied as a reference due to the lack of water source
tracer data. A previous study of snow cover and runoff modelling work in the same basin (Zhang et al.,
2015) provided a potential reference. That work indicated that the contribution of rainfall, snowmelt and
glacier melt in 2006 were 30%, 10% and 60%, respectively, which was markedly different from the result
of this study. The runoff simulation in Zhang et al. (2015) was conducted by a simplified conceptual
model with limited physical mechanism, which did not consider the processes of subsurface runoff and
evaporation. And the glacier melt runoff coefficient (the ratio of glacier melt runoff to the total glacier
melt) estimated by that study was very small (0.182), indicating that a large proportion of glacier melt
did not contribute to the surface runoff directly, which is inconsistent with the common assumption in
previous studies (e.g., Seibert et al., 2018; Schaefli et al., 2005). The extremely low glacier melt runoff
coefficient might lead to overestimation of the contribution of glacier melt. The significant differences
between the two studies mainly resulted from the difference of model structure. Intensive source water
sampling together with systematic glacier observation might improve the behavior of hydrological model
in glacierized basins and help us better understand the runoff processes.
In glacierized basins where glacier meltwater played an important role on runoff generation, the
object of the three MTT estimation methods were different. The total runoff could be divided into
precipitation-triggered runoff (including rainfall-runoff and snowfall-snowmelt-runoff) and glacier melt
runoff. Considering that glacier was also formed by the precipitation over past years, the lumped SW
method should have reflected both runoff processes, because it was based on the tracer data of
precipitation and total runoff. The two model-based methods mainly focused on the precipitation-
triggered runoff, because the glacier revolution process was simplified in the model. The MTT estimation
of total runoff should be based on the assumed MTT of glacier melt water. In this study, assuming the
MTT of glacier melt as 1000 days, the model-based results were similar with SW method, indicating that
the assumption of glacier melt MTT was appropriate, which was actually misleading. The time scale of
glacier update was much longer than this assumed value, because glacier generally took decades to
hundreds of years to move from accumulation zone to ablation zone (Soncini et al., 2014; Yao et al.,
2012). The good agreement among the three methods indicate that the SW method significantly
underestimated the age of glacier. This was mainly due to the limited applicable time scale of stable
isotope in water. It was reported that seasonal cycle of stable isotope in precipitation were most useful
for inferring relatively short travel time of 2-4 years (McGuire and McDonnell, 2006; Sprenger et al.,
2019; Stewart et al., 2010). The assumed glacier melt MTT of 1000 days was within this range, thus the
similar result of three methods could verify the model representation of the precipitation-triggered runoff



process, and the cross validation between MM and FT methods further enhanced the robustness of the
travel time estimation. Consequently, we can expect that if a tracer suitable for longer travel time (e.g.,
$^{14}$C) was used to estimate the proper MTT of total runoff, we could better infer the age of water according
to the model-based estimation of precipitation-triggered runoff MTT.

**5. Conclusions**

A tracer module was integrated into the THREW hydrological model to constrain the various runoff
processes, and was tested in a glacierized catchment on the Tibetan Plateau. Measurements of oxygen
stable isotopes of the stream water were used to calibrate the model parameters, in addition to the
observations of discharge and MODIS SCA. The behaviors of the model, especially the quantifications
of runoff components were compared among the calibration variants with different objective, to test the
value of isotope data on constraining the model parameters. A lumped method (SW) and two model-
based methods (MM and FT) were applied to estimate the water travel time in the study basin. Our main
findings are:
(1) The THREW-t model performed well on simultaneously reproducing the variations of discharge,
snow cover area, and the isotopic composition of stream water, despite of a small water sample number
of precipitation was available to provide isotope input data.
(2) The contributions of rainfall, snowmelt and glacier melt to the annual runoff were quantified as
47.4%, 23.4% and 29.2%. Surface runoff (contributing around 64.9%) was more dominant than
subsurface flow in the annual runoff. Calibration with isotope data significantly reduced the uncertainties
by regulating the competition between rapid and slow runoff components to fit the variation of observed
isotope signature, and resulted in more plausible quantifications of contributions of runoff components
to seasonal runoff.
(3) The estimated MTT of model-based methods MM and FT met well with that of a sin-wave
lumped parameter method, indicating the robustness of travel time estimation benefiting from the use of
water isotope data. The precipitation-triggered MTT was significantly shorter than the MTT of total
runoff, indicating the effect of old glacier meltwater. The MRT was longer than precipitation-triggered
MTT, indicating the catchment scale incomplete mixing processes, and the affinity for young water due
to the rapid flow pathways such as runoff on impermeable glacier surface.

**Code/Data availability**

The isotope data and the code of THREW-t model used in this study are available by contacting the
authors.

**Author contribution**

YN, ZH and FT conceived the idea; LT provided the observation data; FT provided financial support;
YN and conducted analysis; LT and LS provided comments on the analysis; all the authors contributed
to writing and revisions.

**Competing interests**

The authors declare that they have no conflict of interest.



**Acknowledgements**
This study was supported by the National Science Foundation of China (92047301, 91647205). The
authors would like to thank Kunbiao Li from Tsinghua University for the contribution of the coding of
calibration program. The authors thank all the organizations and scientists for the contribution of data
used in this work. Datasets of glacier, snow cover and vegetation for this study are available in these in-
text data citation referees: Liu (2012), Hall and Riggs (2016), Didan (2015) and Myneni et al. (2015).
The digital elevation model (DEM) data set is available at Geospatial Data Cloud site, Computer Network
Information Center, Chinese Academy of Sciences (http://www.gscloud.cn). The meteorological data is
available at China Meteorological Data System (http://data.cma.cn). The soil data is available at the Food
and Agriculture Organization of the United Nations (http://www.fao.org/geonetwork/). All the data used
in this study will be available at the Zenodo website at the time of publication or on request from the
corresponding author (tianfq@mail.tsinghua.edu.cn).
**Financial support**
This study was supported by the National Science Foundation of China (grant no. 92047301, 91647205).

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

fact and hydrological effect of multiphase water transformation in cold regions of the western china:
a review. EARTH ENCE REVIEWS.






**List of figures**



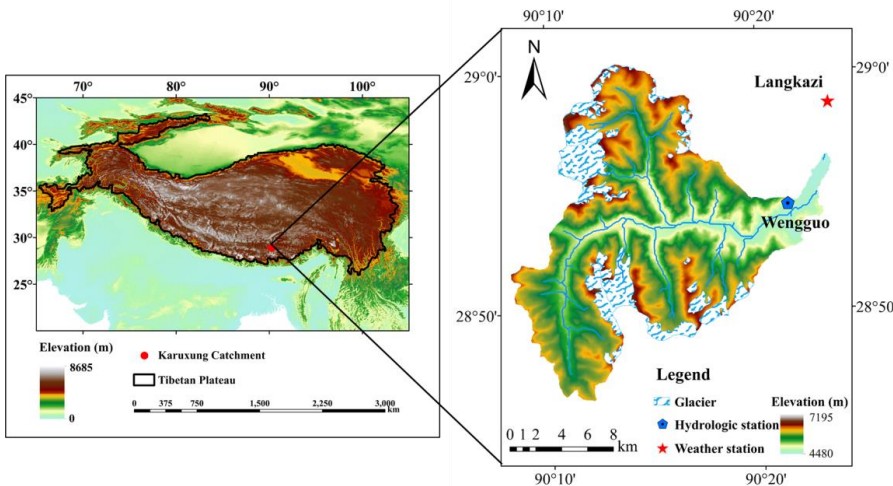


**Figure 1.** Location and topography of the study area






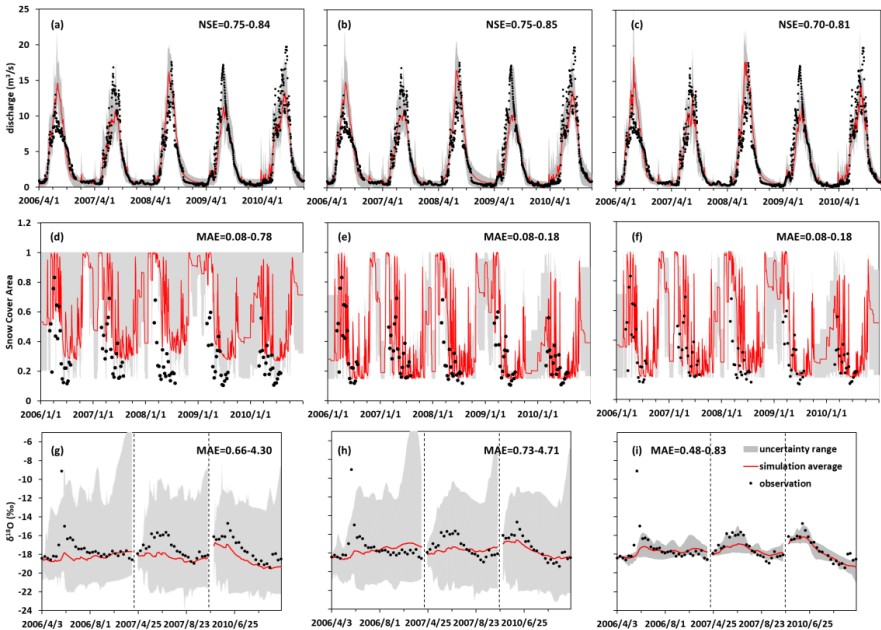


**Figure 2.** Uncertainty ranges of simulations in the calibration period produced by the behavioral parameter sets of the single-objective (subfigure a to c), dual-objective (subfigure d to f) and triple-objective (subfigure g to i) calibration variants.




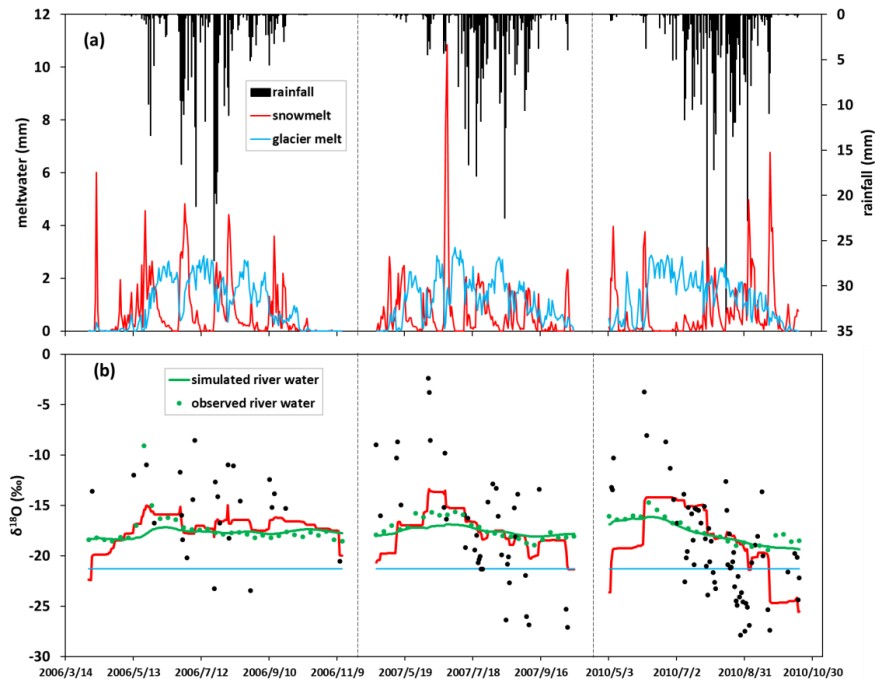


**Figure 3.** Daily simulations of (a) each water source and (b) the corresponding isotopic compositions





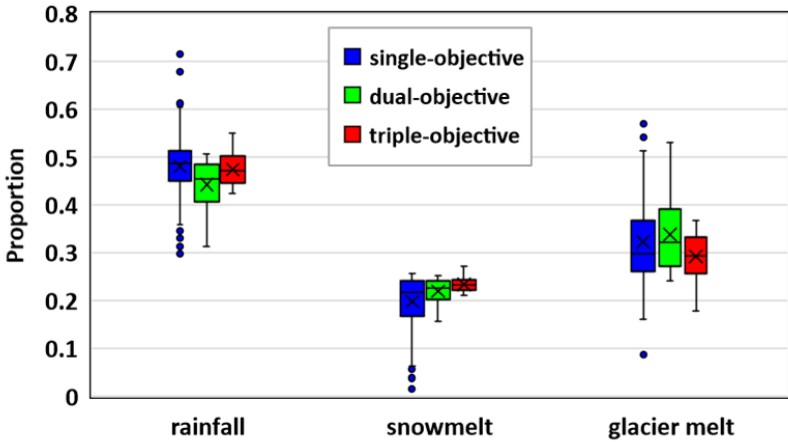


**Figure 4.** Average proportion of different sources in the annual water input for runoff generation



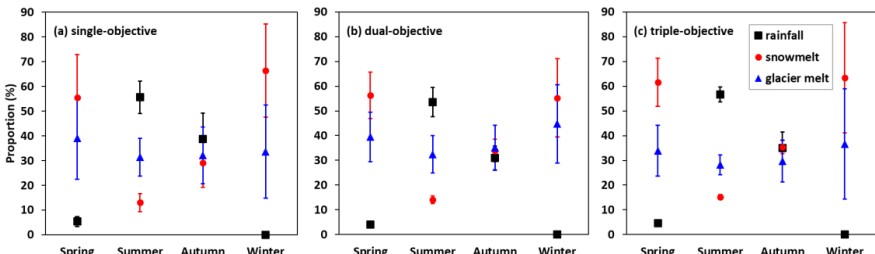

**Figure 5.** Seasonal contributions of rainfall, snowmelt and glacier melt to total water input estimated by
the (a) single-objective, (b) dual-objective and (c) triple-objective calibration variants. The error bars
indicate the uncertainty ranges simulated by the corresponding behavior parameter sets.





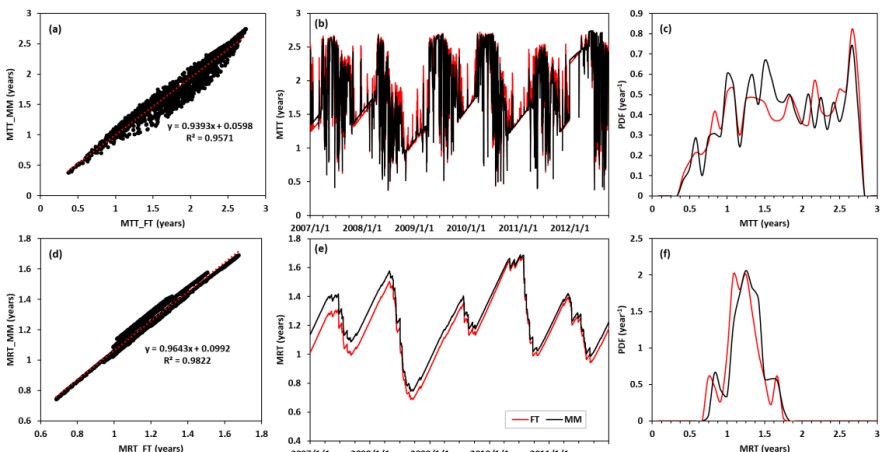


**Figure 6.** Comparison between the MM and FT methods: scatterplots for daily (a) MTT and (d) MRT;
time series of the daily (b) MTT and (e) MRT; and probability density functions of the daily (c) MTT and
(f) MRT.




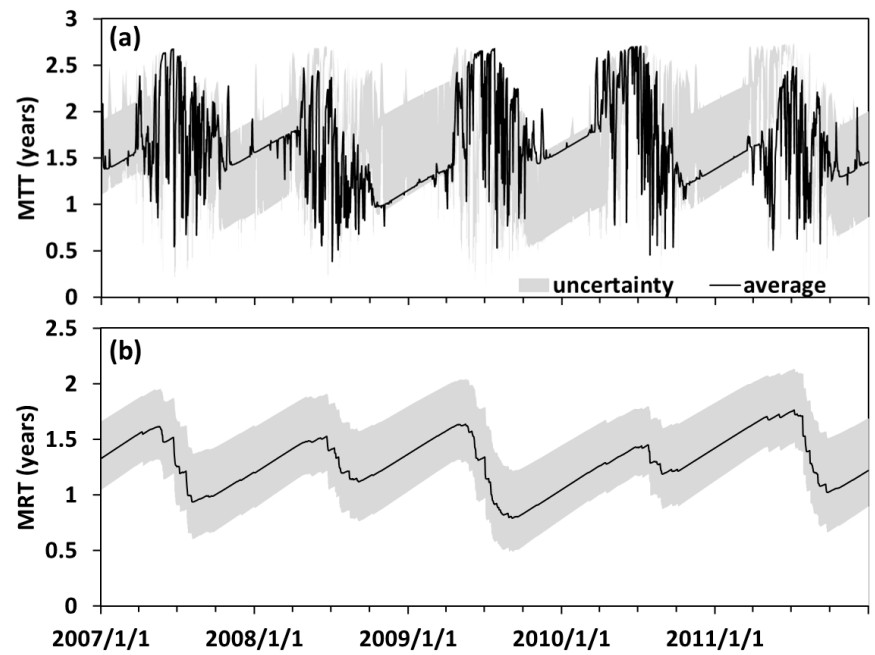


**Figure 7.** Uncertainty ranges of time series (a) MTT and (b) MRT simulated by MM method.



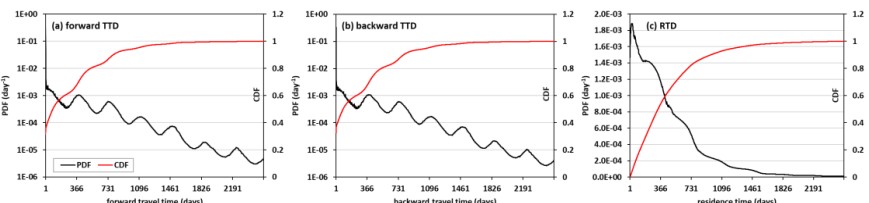

**Figure 8.** The weighted average probability distributions of (a) forward TTD, (b) backward TTD, and (c) RTD estimated by FT method

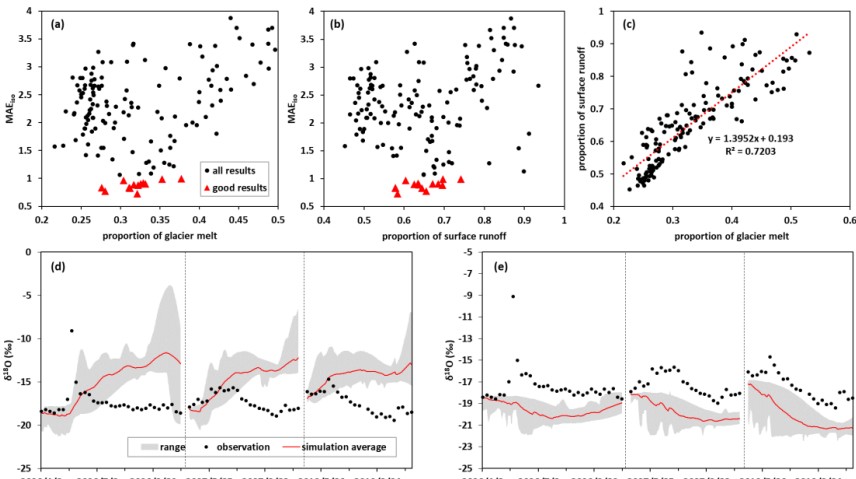

**Figure 9.** The role of isotope calibration on constraining the proportion of runoff components. (a) The relationship between MAE$_{iso}$ and proportion of glacier melt. (b) The relationship between MAE$_{iso}$ and proportion of surface runoff. (c) The relationship between proportion of surface runoff and that of glacier melt. (d) The simulated isotope in stream water produced by the parameter sets estimating proportion of glacier melt higher than 40%. (e) The simulated isotope in stream water produced by the parameter sets estimating proportion of surface runoff lower than 45%.





945 **List of tables**




**Table 1.** Characteristics of precipitation and stream samples

| Year | Period | Precipitation sample number | Stream sample number |
|------|--------|-----------------------------|----------------------|
| 2006 | April 6th to November 11th | 24 | 31 |
| 2007 | April 23rd to October 9th | 39 | 25 |
| 2010 | May 5th to October 18th | 63 | 23 |
| 2011 | March 28th to November 6th | 69 | 32 |
| 2012 | June 16th to September 22nd | 42 | 14 |






**Table 2.** Calibrated parameters of the THREW-t model

| Symbol | Unit | Physical descriptions | Range |
|---|---|---|---|
| $nt$ | - | Manning roughness coefficient for hillslope | 0-0.2 |
| $WM$ | cm | Tension water storage capacity, used in Xinanjiang model (Zhao, 1992) to calculate saturation area | 0-10 |
| $B$ | - | Shape coefficient used in Xinanjiang model to calculate saturation area | 0-1 |
| $KKA$ | - | Coefficient to calculate subsurface runoff in $Rg = KKD \cdot S \cdot K^S_S \cdot (y_S/Z)^{KKA}$, where $S$ is the topographic slope, $K^S_S$ is the saturated hydraulic conductivity, $y_s$ is the depth of saturated groundwater, $Z$ is the total soil depth | 0-6 |
| $KKD$ | - | See description for $KKA$ | 0-0.5 |
| $T_0$ | °C | Melting threshold temperature used in Eqs. (1) and (2) | -5-5 |
| $DDF_N$ | mm/°C/day | Degree day factor for snow | 0-10 |
| $DDF_G$ | mm/°C/day | Degree day factor for glacier | 0-10 |
| $C1$ | - | Coefficient to calculate the runoff concentration process using Muskingum method: $O_2 = C_1 \cdot I_1 + C_2 \cdot I_2 + C_3 \cdot O_1 + C_4 \cdot Q_{lat}$, where $I_1$ and $O_1$ is the inflow and outflow at prior step, $I_2$ and $O_2$ is the inflow and outflow at current step, $Q_{lat}$ is lateral flow of the river channel, $C_3 = 1 - C_1 - C_2$, $C_4 = C_1 + C_2$ | 0-1 |
| $C2$ | - | See description for $C1$ | 0-1 |






**Table 3.** Comparisons of the model performance produced by three calibration variants.

| calibration variant | Number of behavior parameter sets | period | NSE$_{dis}$ [a] | MAE$_{SCA}$ | MAE$_{iso}$ |
|---|---|---|---|---|---|
| Single-objective | 126 | calibration | 0.79 (0.75-0.84) | 0.25 (0.08-0.78) | 2.21 (0.66-4.10) |
| | | validation | 0.79 (0.71-0.84) | 0.24 (0.07-0.79) | 2.53 (0.77-4.88) |
| Dual-objective | 117 | calibration | 0.79 (0.75-0.85) | 0.10 (0.08-0.18) | 2.18 (0.73-4.71) |
| | | validation | 0.80 (0.73-0.84) | 0.08 (0.06-0.19) | 2.38 (0.84-4.96) |
| Triple-objective | 19 | calibration | 0.74 (0.70-0.81) | 0.13 (0.08-0.18) | 0.68 (0.48-0.83) |
| | | validation | 0.79 (0.73-0.84) | 0.11 (0.06-0.18) | 0.93 (0.72-1.19) |

a: Bracketed values represent the minimal and maximal values produced by the behavioral parameter
sets.





**Table 4.** Average percentages of water sources in the annual water input for runoff generation.

|  | Single-objective | Dual-objective | Triple-objective |
|---|---|---|---|
| Rainfall | 48.0 | 44.2 | 47.4 |
| Snow melt | 19.7 | 22.0 | 23.4 |
| Glacier melt | 32.2 | 33.8 | 29.2 |
| Uncertainty [a] | 12.4 | 9.4 | 6.2 |

a: The uncertainty of the contribution is defined as $E = \sqrt{E_R{}^2 + E_N{}^2 + E_G{}^2}$, where $E_R$, $E_N$ and $E_G$
represent the standard deviations of the contributions of the water sources produced by the corresponding
behavioral parameter sets. Subscripts of $R$, $N$ and $G$ represent rainfall, snow meltwater and glacier
meltwater, respectively.





**Table 5.** Simulated contributions of runoff components to annual runoff

|  | Single-objective | Dual-objective | Triple-objective |
|---|---|---|---|
| Surface runoff | 65.9% | 66.4% | 64.9% |
| Subsurface runoff | 34.1% | 33.6% | 35.1% |
| Uncertainty [a] | 11.8% | 12.1% | 4.1% |

a: The uncertainties of the contributions were calculated in the same way in Table 4.