# Peer review of "The value of water isotope data on improving process"

_Hydrology and Earth System Sciences, 2021_

## Author Response (AR1)

Response to the reviews

We would like to express our thanks to the editor and four reviewers for the comments and suggestions on our manuscript. We have revised the manuscript thoroughly based on those comments. address them below on a point-by-point basis. The reviewer's comments are enumerated, and our replies to each comment start with "Response". We look forward to hearing from you.

Yours sincerely,
Yi Nan

The revisions in the manuscript are as followed:

1. Revised the introduction section by adding a general need of isotope at the beginning.

2. Clarified the difference between the snowmelt and glacier melt defined in this study (Line 184-187).

3. Revised the Tables 4 and 5 to list the seasonal contributions of runoff component, and discussed the seasonal characteristic of the runoff component contribution in the main text (Line 438-444; Line 452-456).

4. Added a section and Fig. 10 to discuss the estimated MTT and MRT and their influence factors.

5. Discussed the influence of calibration objective on the result in the limitation section.

6. Simplified the abstract and conclusion parts.

7. Revised all the minor issues addressed by the reviewers.

**Response to Editor**

**Comment 1:** I think the reviewers' comments can be addressed in a modest revision. Additionally, I think the value of this study could be further improved by re-organizing the Introduction section slightly. For example, it would be better to start the manuscript with a general need and some isotope background, rather than from discussing a specific geographic region.

**Response:** Thanks for your comments and suggestions. We have revised the manuscript thoroughly according to the comments from the reviewers. Also, we re-organized the introduction section by starting with a general need of isotope.

**Response to Reviewer #1**

**Comment 1:** Water isotope data have long been recognized to have potential value to hydrological process understanding. However, its potentials have seldom been explored in high mountainous areas, where proper understanding on the complicated hydrology is of critical importance for future projection on hydrological conditions. This paper tries to investigate the value of water isotope data on improving process understanding in a glacierized catchment on the Tibetan Plateau. The authors developed a tracer-aided hydrological model and adopted a long time series of water isotope data to feed the model to quantify the runoff components and MTT/MRT. The results shows that the developed tracer-aided hydrological can substantially reduce the simulation uncertainty with the aid of precipitation and streamflow water isotope data. At the same time, the method adopted in the paper can estimate the MTT/MRT in a reasonable accuracy, which opens a new window to understand the hydrological processes in

the plateau area. The paper is well structured and the language is well written. I strongly recommend its publication on HESS after properly addressing the following comments.

**Response:** Thank you very much for your comment. We have revised the manuscript according to your suggestions.

**Comment 2:** MTT and MRT are useful terms to understand hydrological processes in a catchment. The authors discuss the values of MTT/MRT. More discussions on the influencing factors are encouraged.

**Response:** Thanks for your suggestion. We analyzed the influence of the meteorological factors and wetness condition on the MTT and MRT. We found that the MRT is controlled by the soil water content. The backward MTT is controlled by both soil water content and the precipitation amount during dry season and wet season, respectively. The forward MTT has a strong correlation with temperature, which controls the fraction of snowfall and the rate of evaporation. We have added 4.2 section (Insight from MTT and MRT estimation) to discuss about this in the revision version.

**Comment 3:** The seasonal contribution of water sources is an important result, but the values are quite xxx to identified from Fig. 5, better to show the result by a table. It would also be better to show the result of seasonal contribution of surface and subsurface runoff to better understand the seasonal runoff regimes.

**Response:** Thanks for your suggestion. We have presented the seasonal contribution of water sources and runoff component in Table 4 and 5, and described the characteristic of seasonal contribution of runoff component (surface and subsurface runoff) in the 3.2 section (Line 452-456).

**Comment 4:** In the multiple-objective calibrations, the NSEdis and MAESCA, MAEiso were added directly. Better to clarify the influence of calibration objective function on the result.

**Response:** Thanks for your comment. This is indeed an important issue, and the weight of each calibration objective should be carefully determined when developing a general calibration strategy. But this study aims to illustrate the benefit from the calibration of isotope, and putting the three objective functions together is used to demonstrate that sound simulation for the three objectives can be produced simultaneously. Our result showed that when three objectives were all simulated well, the uncertainty of parameter and runoff component contribution was significantly reduced compared to the condition when only one objective was satisfied. To this end, the influence of weight of each objective is beyond the scope of this study although it is an important topic. We have clarified this in the limitation section (Line 635-647).

**Comment 5:** The abstract must be concise, summaries the research aims, methodology, results and discussion, and conclusions. This lacks in your abstract.

**Response:** Thanks for your suggestion. We have revised the abstract to make it more concise and contain all the important things in aims, methods, results and discussion.

**Comment 6:** The conclusions should be brief and more informative. Please cut down the conclusion to a short paragraph.

**Response:** Thanks for your suggestion. We have simplified the conclusion part to one paragraph.

**Comment 7:** May consider the subscribe "N" to "S" to represent snow for quicker understanding.

**Response:** Thanks for your suggestion. We have changed the subscribe of snow to "S" in the revision version.

**Response to Reviewer #2**

**Comment 1:** This paper is very interesting by coupling the isotopic tracers and hydrological model, which has the potential to solve the problem of hydrograph separation in a large-scale catchment. The authors get reasonable results with their model. So I accept the paper after a minor revision.

**Response:**

Thank you very much for your comment. We have revised the manuscript according to your suggestions.

**Comment 2:** Could the authors add some precious work to compare with their results on the hydrograph separation as well as the MRT and MTT in the discussion section?

**Response:**

Thanks for your suggestion. The result of hydrograph separation has been compared with another work conducted in the same catchment (Zhang et al., 2015) in the limitation section. We have additionally added section 4.2 to discuss the MTT and MRT in two aspects. One is to compare the estimated MTT and MRT with other studies (Ala-aho et al., 2017) conducted in snow-influenced catchments, and to explain the reason for the differences. We found that the MTT and MRT significantly distinguished among snow-influenced catchments, which was mainly attributed to the differences in topography and soil characteristics. The other is to analyze the influence factor of MTT and MRT, and compare the result with previous studies.

We found that the MRT is controlled by the soil water content. The backward MTT is controlled by both soil water content and the precipitation amount during dry season and wet season, respectively. The forward MTT has a strong correlation with temperature, which controls the fraction of snowfall and the rate of evaporation. These findings are similar with the results reported by Heidbuechel et al. (2012), McMillan et al. (2012) and Hrachowitz et al. (2013), which estimated the MTT using different methods from our study.

**Comment 3:** Page 2: Method: how to define the snow-melt and glacier-melt? And how to obtain their isotope values? In my view, it is really hard to differentiate them in the filed work because they are always mixed when do the sampling.

**Response:**

Thanks for your question. As a work focusing on quantifying the contribution of water sources, the simulation of water sources and their isotope composition is indeed need clarify.

The snowmelt and glacier melt were mainly differentiated according to the glacier coverage from the Inventory data. The melt water in glacier covered region is glacier melt, and the melt water in non-glacier region is snow melt. The two kinds of water sources are melting with different DDF. We simulated the variation of snow cover area using the method described in 2.2. For model simplification, the evolution of glacier thickness and area was not simulated.

As for the isotope values, the snowpack was regarded similarly with other hydrological simulation units, thus the isotope composition was simulated similarly by Eq. 8. The isotope composition of glacier meltwater was assumed to be constant, adopting the value reported in published paper (Gao et al., 2009).

We have clarified the above issues in the Method section (Line 184-187 and 237-238) in the revision version.

**Comment 4:** Page 10 Lines 353 -360 The calibration is quite interesting. The finding 'The single-objective calibration produced good performance for the simulation of discharge, but had an extremely poor performance for the simulations of SCA and $\delta 18O$' means without the tracers, even the calibration is accepted, the model may still bring large uncertainty. Is my understanding correct?

**Response:**

Yes, you are right. According to our result, even when the model produces good discharge simulation with high NSEdis, the internal processes can be very different, because discharge is only an external characteristic of catchment. This phenomenon has also been highlighted in some previous modelling works (such as Birkel et al. 2011, Campell et al. 2012, Chen et al. 2017). This indicates that the parameter cannot be constrained well solely by the behavior of discharge simulation. Consequently, involving more datasets such as snow, glacier, isotope is

helpful for reducing the parameter equifinality.

**Comment 5:** Page 11 Lines 391-393 Please reconsider your explanation on the river O-18. The temperature effect is kind of a statistical result, while the effect of southwest monsoon is more likely a reason to cause the temperature effect, and thus it is not suitable to put them together.
**Response:**
Thanks for your suggestion. We will change the explanation by attributing the reduced $\delta^{18}O$ to the effect of monsoon in the revision version.

**Comment 6:** Figure 3 Black circles and red line in sub-figure b are same to Fig.2?
**Response:**
No. They are same to sub-figure a of Fig. 3 (i.e., the black circles and red line represent the isotope composition of rainfall and snowmelt). We have clarified this in the revised version.

**References:**

Ala-Aho, P. , Tetzlaff, D. , Mcnamara, J. P. , Laudon, H. , & Soulsby, C. . (2017). Using isotopes to constrain water flux and age estimates in snow-influenced catchments using the starr (spatially distributed tracer-aided rainfall–runoff) model. Hydrology and Earth System ences Discussions, 21(10), 5089-5110.

Birkel, C., Tetzlaff, D., Dunn, S. M., & Soulsby, C. (2011). Using time domain and geographic source tracers to conceptualize streamflow generation processes in lumped rainfall‐runoff models. Water Resources Research, 47(2).

Capell, R., Tetzlaff, D., & Soulsby, C. (2012). Can time domain and source area tracers reduce uncertainty in rainfall‐runoff models in larger heterogeneous catchments?. Water Resources Research, 48(9).

Chen, X. , Long, D. , Hong, Y. , Zeng, C. , & Yan, D. . (2017). Improved modeling of snow and glacier melting by a progressive two-stage calibration strategy with grace and multisource data: how snow and glacier meltwater contributes to the runoff of the upper brahmaputra river basin?. Water resources research, 53(3), 2431-2466.

Gao J. , Tian L. , & Liu Y. (2009). Oxygen isotope variation in the water cycle of the Yamdrok-tso Lake Basin in southern Tibetan Plateau. Chinese Sci Bull, 54: 2758—2765

Heidbuechel, I. ,  Troch, P. A. ,  Lyon, S. W. , &  Weiler, M. . (2012). The master transit time distribution of variable flow systems. Water Resources Research, 48(6), 6520.

Hrachowitz, M. , Savenije, H. , Bogaard, T. A. , Tetzlaff, D. , & Soulsby, C. . (2013). What can

flux tracking teach us about water age distribution patterns and their temporal dynamics?. Hydrology and Earth System Sciences, 17(2), 533-564.

McMillan, H., Tetzlaff, D., Clark, M., Soulsby, C., 2012. Do time-variable tracers aid the evaluation of hydrological model structure? A multimodel approach. Water Resour. Res. 48. https://doi.org/10.1029/2011WR011688

Zhang, F., Zhang, H., Hagen, S. C., Ye, M., Wang, D., Gui, D., ... & Liu, J. (2015). Snow cover and runoff modelling in a high mountain catchment with scarce data: effects of temperature and precipitation parameters. Hydrological processes, 29(1), 52-65.